# RP-UHPLC-MS Chemical Profiling, Biological and In Silico Docking Studies to Unravel the Therapeutic Potential of *Heliotropium crispum* Desf. as a Novel Source of Neuroprotective Bioactive Compounds

**DOI:** 10.3390/biom11010053

**Published:** 2021-01-04

**Authors:** Adeel Arshad, Saeed Ahemad, Hammad Saleem, Muhammad Saleem, Gokhan Zengin, Hassan H. Abdallah, Muhammad Imran Tousif, Nafees Ahemad, Mohamad Fawzi Mahomoodally

**Affiliations:** 1Department of Pharmacy, Faculty of Pharmacy and Alternative Medicine, The Islamia University of Bahawalpur, Bahawalpur 63100, Pakistan; adeelarshad.11@gmail.com (A.A.); rsahmed_iub@yahoo.com (S.A.); 2School of Pharmacy, Monash University Malaysia, Jalan Lagoon Selatan, Bandar Sunway 47500, Selangor Darul Ehsan, Malaysia; nafees.ahemad@monash.edu (H.S.); hammad.saleem@monash.edu (N.A.); 3Institute of Pharmaceutical Sciences (IPS), University of Veterinary & Animal Sciences (UVAS), Lahore 54000, Pakistan; 4Department of Chemistry, The Islamia University of Bahawalpur, Bahawalpur 63100, Pakistan; m.saleem@iub.edu.pk; 5Department of Biology, Science Faculty, Selcuk University, 42130 Konya, Turkey; hammad.saleem@uvas.edu.pk; 6Chemistry Department, College of Education, Salahaddin University-Erbil, Erbil 44001, Iraq; Hassan.abdullah@su.edu.krd; 7Department of Chemistry, Township Campus, University of Education Lahore, Lahore 54000, Pakistan; gokhanzengin@selcuk.edu.tr; 8Department for Management of Science and Technology Development, Ton Duc Thang University, Ho Chi Minh City 70000, Vietnam; 9Faculty of Applied Sciences, Ton Duc Thang University, Ho Chi Minh City 70000, Vietnam

**Keywords:** *Heliotropium crispum*, methanolic extract, antioxidant, enzyme inhibition, docking, phytochemical profiling, RP-UHPLC-MS

## Abstract

*Heliotropium* is one of the most important plant genera to have conventional folklore importance, and hence is a potential source of bioactive compounds. Thus, the present study was designed to explore the therapeutic potential of *Heliotropium crispum* Desf., a relatively under-explored medicinal plant species. Methanolic extracts prepared from a whole plant of *H. crispum* were studied for phytochemical composition and possible in vitro and in silico biological properties. Antioxidant potential was assessed via six different assays, and enzyme inhibition potential against key clinical enzymes involved in neurodegenerative diseases (acetylcholinesterase (AChE) and butyrylcholinesterase (BChE)), diabetes (α-amylase and α-glucosidase), and skin problems (tyrosinase) was assayed. Phytochemical composition was established via determination of the total bioactive contents and reverse phase ultra-high performance liquid chromatography mass spectrometry (RP-UHPLC-MS) analysis. Chemical profiling revealed the tentative presence of 50 secondary metabolites. The plant extract exhibited significant inhibition against AChE and BChE enzymes, with values of 3.80 and 3.44 mg GALAE/g extract, respectively. Further, the extract displayed considerable free radical scavenging activity against DPPH and ABTS radicals, with potential values of 43.19 and 41.80 mg TE/g extract, respectively. In addition, the selected compounds were then docked against the tested enzymes, which have shown high inhibition affinity. To conclude, *H. crispum* was found to harbor bioactive compounds and showed potent biological activities which could be further explored for potential uses in nutraceutical and pharmaceutical industries, particularly as a neuroprotective agent.

## 1. Introduction

Plants are an integral part of traditional medicines, with a huge variety of bioactive components that are effective against various diseases. A wide variety of medicinally active compounds harbored within the plant kingdom are synthesized by the plants through multiple metabolic pathways, while plant biological activities are attributed to their metabolites [1]. Thus, there is a need to develop fast and reliable methods for the screening of plant extracts [2]. For the past few decades, the importance of medicinal plants for treating various infections has tremendously increased because of the fact that a large number of people belonging to different populations depend upon the use of phytomedicines, due to the unavailability of primary healthcare facilities [3]. According to World Health Organization reports on phytomedicines, more than 25% of drugs which have been prescribed in recent years are obtained from different plant sources [4].

*Heliotropium,* comprising ~250–300 species, is among the largest genera of the plant family Boraginaceae. They are distributed throughout the world; however, only about 23 species are growing in various regions of Pakistan [5]. The word “heliotrope” has been specified for plants that have the property of turning their leaves in the direction of the sun [6]. *Heliotropium* is famous for its medicinal use in various countries such as Malaysia, Mauritius, and Tanzania. For example, in Malaysia, *H. indicum* paste has been used against pyoderma, putrefaction, and ringworm infection [7]. *H. strigosum* juice is effective for eye sores and against snake bites, while its powdered form has been reported as an effective treatment in rheumatic arthritis. Furthermore, laxative, diuretic, blood cleanser, and jaundice treatment has been achieved through a decoction of *H. strigosum* [8]. Leaf juice of *H. dasycarpum* has useful coagulating properties to stop wound bleeding and to treat throat infection in Tanzania [8]. In Mauritius, leaves of *H. indicum* are used against urinary tract infections, rheumatism, and small wounds [9]. This plant has been documented to contain the important anticancer compound indicine-*N*-oxide, which shows significant activity against P388 leukemia. It is also active against B16 melanoma, and the aqueous, ethanol, and acidic extracts of the plant are reported as antifungal [9]. *H. crispium*, growing in the Lal-suhanra forest of the Punjab in Pakistan, is commonly known as Kharsan. Its powdered dry extract, along with candy and water, is used as a cooling agent. It is used for increasing the lactation of animals [9,10,11]. Silver nanoparticles of aqueous plant extract have shown excellent potential in different drug resistance bacteria [12]. The extensive collection of medicinal plants from this genus and their potential uses in the local medicinal system prompt us to investigate *H. crispium* for its detailed biological activity. Chemical profiling of the methanolic extract was done to establish a possible link between the bioactive contents and their biological activity. Various assays (DPPH, ABTS, FRAP, CUPRAC, phosphomolybdenum, and metal chelation assays) were performed to dig out the antioxidant potential of the extract. In addition, the inhibitory potential of the extract against the key enzymes responsible for world-alarming diseases like neurodegenerative disorder (acetylcholinesterase (AChE) and butyrylcholinesterase (BChE)), diabetes (α-glucosidase and α-amylase), and skin problems (tyrosinase) were estimated. Moreover, in silico docking studies were also performed.

## 2. Results and Discussion

### 2.1. Total Bioactive Contents

Recently, researchers have shown a renewed interest in the investigation of the bioactivities of phenolics, especially in flavonoids from higher plants, due to there being fewer or no side effects resulting from their use and that they can be taken as major dietary constituents [13]. In the present study, a methanolic extract of *H. crispum* showed high total phenolic contents (24.84 mg GAE/g extract) compared to the flavonoid contents (19.73 mg GAE/g extract), as presented in Table 1. In a published report on the outcome of a spectrophotometric assay of *H. strigosum*, the methanol extract showed a higher amount of total phenolic contents than the aqueous, ethyl acetate, chloroform, and *n*-hexane fractions [14]. Likewise, another study conducted on different extracts of *H. indicum* showed a higher amount of phenolic contents in a hydro-alcoholic crude extract compared with the methanolic, ethyl acetate, and hexane fractions [15]. The reported findings support our results for the presence of higher phenolic contents in a methanolic extract.

### 2.2. Identification of Secondary Metabolites by Reverse Phase Ultra-High Performance Liquid Chromatography Mass Spectrometry (RP-UHPLC-QTOF-MS)

The methanolic extract of *H. crispum* was subjected to a UHPLC-MS (positive and negative ionization mode) screening, which resulted in the possible identification of 50 secondary metabolites (Table 2 and Table 3). These secondary metabolites belong to different classes such as irodide, pyrrolizidine alkaloids, peptides, pyruvic and coumaric acids, polyphenolic compounds, and long-chain fatty acids. Among the irodides identified through the negative mode, nepetaside (**1**) was found to be a known phytochemical [16], while the unknown compound **2** was tentatively identified as a derivative of **1** (Scheme 1). UHPLC-MS analysis depicted the molecular formula of compound **1** as C_16_H_26_O_8_ with four double bond equivalents (DBE), while the unknown compound **2** (C_17_H_30_O_8_ with three DBE) was elucidated as though the carbonyl moiety of the ester function might have been reduced to an ether linkage. Further, the mass analysis showed a difference of 16 amu between **1** and **2,** which also confirmed the methylation along with a reduction of **1** into **2**. This kind of methylation is a common biosynthetic process found in natural products, specifically in irodide classes [16].

The tentatively identified plant pyrrolizidine alkaloids (in positive ionization mode) were supinine (**3**), europine **(4)**, and heliotrine **(5)** [14]. These pyrrolizidine alkaloids (Scheme 2) are known phytochemicals from various *Heliotropium* species. Pyrrolizidine alkaloid **6** was found to be a new derivative of **4** (C_16_H_27_NO_6_), since the UHPLC-MS data displayed molecular ions for **4** and **6** at *m/z* 329.1847 (C_16_H_27_NO_6_) and 331.2003 (C_16_H_29_NO_6_), respectively, which indicated that compound **6** is a reduced form of **4**. Thus, on the basis of the molecular formula and the number of DBE, two structures, **6a** and **6b** are possible; however, structure **6a** is more suitable based on its stability factor and the fact that both saturated and unsaturated pyrrolidine rings are reported in pyrrolizidine alkaloids from various *Heliotropium* species [5]. Further, a similarity with structure **b** (reduced ester linkage) was not reported in the literature. Therefore, compound **6a** is identified as new pyrrolidine alkaloids form this plant. Further compounds **3, 4, 5,** and **6a** are biosynthetic analogs, as compound **3** is converted into **4** and **5** by simple methylation and hydroxylation, and these reactions are very common in the biosynthesis of alkaloids. Similarly, compound **4** on dehydrogenation gives compound **6a**.

A total of 13 known nitrogen-containing secondary metabolites were tentatively identified through negative ionization mode, including mostly cyclic and acyclic peptides derivatives; 5-acetylamino-6-formylamino-3-methyluracil (**7**), 1-methylxanthine (**8**), and hypoxanthine (**9**) of the xanthine family compounds [17]. *N*-Acryloylglycine (**10**) and orysastrobin (**11**) are strobilurins derivatives [18], which are widely used as a fungicide [19]. Other identified compounds, including mecarbinzid (**12**), the oxidized compound oplophorus luciferin (**13**), and biotin-X-NHS (**14**), are important secondary metabolites. Carbinoxamine (**15**), a natural product derived from carbamic acid, was also found in the extract; however, these compounds are very rare in nature, especially from plant sources. In the past, natural products derived from carbamic acid have been isolated from the Taiwanese plant *Magnolia kachirachirai* [20] and the Pakistani plant *Vincetoxicum stocksii* [21]. The nitrogen-based compounds which were identified through the positive mode of UHPLC-MS analysis include tranexamic acid (**16**), 5-pentyloxazole (**17**), homoarecoline (**18**), *N*-(3-oxododecanoyl) homoserine lactone (**19**), alizapride (**20**), *L*-cladinose (**21**), N1, N10-dicoumaroylspermidine (**22**), 9-acetoxyfukinanolide (**23**), and dodemorph (**24**).

Among other metabolites, nine secondary metabolites that were tentatively identified from the negative ionization mode of screening were from pyruvic acid derivatives such as *β*-hydroxypyruvic acid (**25**) and nonic acid (**26**). Further coumaric acid derivatives include compounds isobergaptene (**27)**, 2-hydroxy-3,4-dimethoxybenzoic acid (**28**), 3-methoxymandelic acid-4-*O*-sulfate (**29**), *p*-salicylic acid (**30**), cis-ferulic acid [arabinosyl-(1→3)-[glucosyl-(1→6)]-glucosyl] ester (**31**), and ferulic acid (**32**). These important phenolic compounds are known for various potent antioxidant and enzyme inhibitory activities [22]. The tentatively identified polyphenolic compounds included 2-hydroxy-3,4-dimethoxy-6-methyl-5-(sulfooxy) (**33**), 1-hexanol arabinosylglucoside (**34**), (S)-chiral alcohol, and (3S)-4-(3-acetyl-5-hydroxy-4-oxo-1,2,3,4-tetra hydronaphthalen-2-yl)-3-hydroxy butanoic acid (**35**). Flavonoids which were identified included cimifugin (**36**), rivenprost (**37**), eupatin 3-*O*-sulfate (**38**), tamadone (**39**), wightin (**40**), salvianolic acid A (**41**), and lithospermic acid (**42**). In addition, eight long-chain fatty acids were also tentatively identified through the negative ionization mode of UHPLC-MS analysis, which included 11-hydroperoxy-12,13-epoxy-9-octadecenoic acid (**43**), 5,8,12-trihydroxy-9-octadecenoic acid (**44**), 9,16-dihydroxy-palmitic acid (**45**), 9,10-epoxy-18-hydroxystearate (**46**), α-9(10)-EpODE (**47)**, 12-oxo-10Z-octadecenoic acid (**48**), 9Z,12Z,15E-octadecatrienoic acid (**49**), and 6E,9E-octadecadienoic acid (**50**).

### 2.3. Antioxidant Assays

In the present investigation, the free radical scavenging assessment of *H. crispum* extract was made through DPPH and ABTS, with values of 43.19 mg TE/g extract and 41.80 mg TE/g extract, respectively. Meanwhile, reducing power potential was achieved through FRAP and CUPRAC assays, with a potential of 63.11 mg TE/g extract and 116.11 mg TE/g extract, respectively (Table 1). The phosphomolybdenum assay, which measured the total antioxidant potential, exhibited 2.11 mmol TE/g extract potential by the tested extract. In addition to the above assays, the metal chelating potential was observed to be 46.72 mg EDTAE/g extract. These results offer a better understanding of the antioxidant potential of *H. crispum*. In the previously reported data [14], the *H. strigosum* methanolic extract also showed higher potential in the DPPH assay, with IC_50_ of 10.45 μg/mL having higher phenolic contents (84.50 ± 0.06 µg QE/mg of plant extract). In another report, the major product of the dichloromethane extract of *H. ovalifolium* has been reported to contain heliotropamide, an alkaloid with a novel oxopyrrolidine-3-carboxamide central moiety, which exhibited radical-scavenging properties in the DPPH test [23]. Similarly, the methanol extract of *H. indicum* presented DPPH inhibition, with IC_50_ of 306 μg/mL [15].

### 2.4. Enzyme Inhibition Potential

Interest is increasing in the use of natural enzyme inhibitors to combat global health problems including Alzheimer’s disease, Diabetes mellitus, hyperpigmentation, and hypertension. The prevalence of these diseases is critically increasing worldwide, and thus, effective strategies are required to control these diseases. With this in mind, the discovery of natural and safe enzyme inhibitors is one of the most investigated subjects in science [24]. Previously, the methanol extract of *H. strigosum* showed an amylase inhibition potential with an IC_50_ value of 9.97 + 0.01 μg/mL. Heliotropamide, a new alkaloid with a novel oxopyrrolidine-3-carboxamide central moiety, has been isolated as the major product of the dichloromethane extract of *H. ovalifolium* aerial parts, which showed inhibitory potential toward acetylcholinesterase [23]. The whole herb of *H. digynum* was tested against a glucosidase enzyme at 25 ppm and showed a weak percentage inhibition [25]. In the present study, we tested the enzyme inhibitory effects of the *H. crispum* extract against cholinesterases, tyrosinase, α-amylase, and α-glucosidase enzymes. The results are illustrated in Table 4. The extract of *H. crispum* possesses the highest activity against AChE (3.80 mg GALAE/g extract) and BChE (3.44 mg GALAE/g extract), while in the α-glucosidase assay, it showed a 1.86 mmol ACAE/g extract inhibitory activity; however, in the α-amylase inhibition assay, the extract was found comparatively less active, with a value of 0.57 mmol ACAE/g extract. The same extract showed the promising tyrosinase inhibition with a 129.65 mg KAE/g extract inhibition potential.

### 2.5. Docking Results

Docking calculations offer essential quantitative results. The binding affinity and the inhibition constant are among these results. As shown in Table 5, the binding affinity or the docking free energy along with the inhibition constants are listed for the eight selected compounds. Interestingly, lithospermic acid (**42**) and salvianolic acid A (**41**) have shown the best binding affinity against the five enzymes which may indicate a potential bioactive role for these compounds. Figure 1 gives a better insight into the interactions of these compounds at the active site and offers explanation for the relative high binding affinity. Strong non-bonding interactions were found between these compounds and the active site, such as hydrogen bonds, π–π interactions, and electrostatic forces.

In order to investigate the interactions of the highest affinity compounds, lithospermic acid (**42**) and salvianolic acid A (**41**) were further studied; the affinity of these compounds against the studied enzymes, along with their interactions with the active site of the corresponding enzymes, are shown in Table 6. Hydrogen bonds and the π–π interactions may represent the strongest non-bonded interactions. Although the highlighted two compounds have shown a similar binding affinity, they may interact with similar or different amino acids at the active site of the enzyme. For example, lithospermic acid (**42**) was involved with Asn298, Asp297, Arg204, Leu232, and Gly230 of the α-amylase enzyme, while in the case of salvianolic acid A (**41**), the formed hydrogen bonds are with Tyr155, Asp206, Asp340, Glu35, and Tyr79 amino acids. This difference is attributed to the difference in the docked conformations of these compounds, as shown in Figure 1. The same two compounds have shown different behavior in the case of AChE, in which they have formed hydrogen bonds with the same amino acids, such as Phe295, Phe383, Tyr124, and Ser293. In the case of BChE, α-glucosidase, and tyrosinase enzymes, these two compounds have interacted with different amino acids. Similarly, the π–π interactions of the selected compounds did not follow a specific trend. For example, both compounds formed a π–π interaction with the Phe601 and Pro201 of the active site of α-glucosidase and tyrosinase enzymes, respectively, and with different amino acids of the rest of the enzymes.

## 3. Materials and Methods 

### 3.1. Plant Material and Extraction

The whole plant material was collected from Cholistan Desert near Baghdad Campus of the Islamia University of Bahawalpur, Pakistan. The material was identified by plant taxonomist Dr. Muhammad Serwar, Department of Life Sciences, the Islamia University of Bahawalpur, where a voucher specimen-2469 was deposited in the herbarium. The 15-day shade-dried whole plant material was ground into 10.0 g of powder, which was then extracted with 100.0 mL methanol in an ultrasonic ice-water bath for 1 h and was filtered. The filtrate was concentrated under vacuum and was transferred to an autosampler vial for UHPLC-MS analysis.

### 3.2. Total Bioactive Contents

Folin–Ciocalteu and AlCl_3_ assays were used for a quantitative estimation of the total phenolics contents (TPC) and total flavonoid contents (TFC) [26] using gallic acid (mg GAEs/g extract) and rutin (mg RE/g extract) as equivalents, respectively. The details of these protocols are provided in the Appendix A.

### 3.3. RP-UHPLC-MS Analysis

Secondary metabolites profiling was done by reverse phase ultra-high performance liquid chromatography mass spectrometry (RP-UHPLC-MS) analysis utilizing an Agilent 1290 Infinity UHPLC system, which was coupled with an Agilent 6520 Accurate-Mass Q-TOF mass spectrometer with a dual ESI source, as reported previously [27]. The details of the RP-UHPLC-MS analysis are provided in the Appendix A.

### 3.4. Antioxidant Assays

Methods reported by Grochowski et al., 2017 [26], were applied to measure the DPPH, ABTS, FRAP, CUPRAC, and total antioxidant capacity (phosphomolybdenum assay), which were expressed as a trolox equivalent, while the metal chelating effects of the extracts were calculated by ethylenediaminetetraacetic acid (EDTA) as reference. The details of these protocols are provided in the Appendix A.

### 3.5. Enzyme Inhibition Assays

In vitro enzyme inhibitory assays (AChE, BChE, tyrosinase, α-amylase, and α-glucosidase), as reported previously [26], were also performed with the methanolic extract. The positive references used were acarbose for α-amylase and α-glucosidase, galantamine for AChE and BChE, and kojic acid for tyrosinase. The details of these protocols are provided in the Appendix A.

### 3.6. Docking Calculations

Eight compounds which were identified as major components, namely, cimifugin **(36)**, europine **(4)**, heliotrine **(5)**, 11-hydroperoxy-12, 13-epoxy-9-octadecenoic acid (**43**), lithospermic acid **(42)**, salvianolic acid A **(41)**, supinine (**3**), and 5, 8, 12-trihydroxy-9-octadecenoic acid (**44**), were selected for the docking study. The target enzymes of these compounds were α-amylase, AChE, BChE, α-glucosidase, and tyrosinase enzymes. The initial 3D structures of these inhibitors were downloaded from the ZINC database [28]. The structures were prepared for the docking calculations through an optimization process using the semi-empirical AM1 method [29] and saved in the format of mol2 files. The crystal structures of the five enzymes were downloaded from RCSB PDB with the following pdb codes: 4EY6 for the AChE enzyme in complex with galantamine; 1P0P for BChE in complex with butyrylthiocholine; 5I38 for tyrosinase with kojic acid; and 7TAA and 3W37, which were used to represent the α-amylase and α-glucosidase enzymes with the acarbose inhibitor, respectively. Water molecules and all the co-crystalized molecules were removed from the protein structure. Autodock 4 (Molinspiration Database) was used to calculate the binding affinity of the eight compounds with the active site of the enzymes where Kollman united atom charges were used to neutralize the protein which was immersed in a grid box of 60 × 60 × 60 dimensions with 0.375 Å distance. A total of 250 conformations were scanned for each inhibitor using a Lamarckian genetic algorithm, and the interactions were analyzed using Discovery studio 5.0 visualizer.

### 3.7. Statistical Analysis

All the experiments were performed in triplicates, while the results are presented as mean values. Statistical analysis was performed using one-way ANOVA and SPSS v. 17.0 software. The R software v. 3.5.1 was used for statistical calculations; the statistics showed a significant *p* value, i.e., <0.05.

## 4. Conclusions

This study is the first report on the chemical composition and biological (in vitro and in silico) activities of *H. crispium*. It can be concluded that the methanol extract showed consistently high total phenolic contents, a number of individual phenolic compounds with previous antioxidant propensities, as well as enzyme inhibitory potential. The RP-UHPLC-QTOF-MS screening showed the presence of important pyrrolizidine alkaloids, peptides, irodide, and phenolic compounds, which are famous for their antioxidant activities; large numbers of these were identified in this plant. Docking studies for the selected compounds have shown a high inhibition activity of lithospermic acid **(42)** and salvianolic acid A **(41)** against the five studied enzymes. The significant enzyme inhibition, docking results, and antioxidant activities of the *H. crispium* plant could pave a path to further exploration of this plant for the isolation of pure compounds responsible for the observed biological properties.

## Data Availability

Not applicable.

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
