# Peer review of "RP-UHPLC-MS Chemical Profiling, Biological and In Silico Docking Studies to Unravel the Therapeutic Potential of Heliotropium crispum Desf. as a Novel Source of Neuroprotective Bioactive Compounds"

_biomolecules, 2021, doi:10.3390/biom11010053_

Round 1
Reviewer 1 Report
Authors investigated the chemical profile and biological activities of H. crispum. Revision is recommended to confirm and improve the research.
- Authors analyzed the chemical profiles and in silico study. Molecular docking suggested the potential but not exact results. Biological activitis of suggested bioactive constituents should be confirmed by experiment.
- Manuscript has too repetitive and too much literatur survey. ex) "In a published report on the outcome of spectrophotometric 93 assay of H. strigosum, methanol extract had higher amount of total phenolic (84.50 ± 0.06 pg GAE/ Molecules 2020, 25, x FOR PEER REVIEW 3 of 14 94 mg of plant extract) contents than aqueous, ethyl acetate, chloroform and n-hexane fractions, 95 respectively [14], whereas, the phenolic contents in hydro-alcoholic crude extract, methanolic, ethyl 96 acetate and hexane fractions of H. indicum has been reported to be 5.49, 2.96, 9.76, and 2.32 mg/g, 97 respectively". Only key points needs to be described.
- Antioxidant activity of this plant was already reported.
- Table 4 needs IC50 for general comparison.
- Plant binominal name should be in italic
- Many typos throughout the manuscript.
Author Response
Reviewer # 1:
Comment: Authors investigated the chemical profile and biological activities of H. crispum. Revision is recommended to confirm and improve the research.
Response: We thanks the reviewers for his/her suggestions. All the changes as indicated by the reviewer have been highlighted yellow in the text.
Comment: Authors analyzed the chemical profiles and in silico study. Molecular docking suggested the potential but not exact results. Biological activitis of suggested bioactive constituents should be confirmed by experiment. Manuscript has too repetitive and too much literatur survey. ex) "In a published report on the outcome of spectrophotometric 93 assay of H. strigosum, methanol extract had higher amount of total phenolic (84.50 ± 0.06 pg GAE/ mg of plant extract) contents than aqueous, ethyl acetate, chloroform and n-hexane fractions, 95 respectively [14], whereas, the phenolic contents in hydro-alcoholic crude extract, methanolic, ethyl 96 acetate and hexane fractions of H. indicum has been reported to be 5.49, 2.96, 9.76, and 2.32 mg/g, 97 respectively". Only key points needs to be described.
Response: We have modified the sentence as requested.
Comment: Antioxidant activity of this plant was already reported.
Response: This study is a recent attempt to provide a comprehensive report on the biological and chemical profile of this plant. The enzymatic assays conducted is unique and the panoply of antioxidants assays make this manuscript apart from previous studies. In addition, in silico modelling is being for key compounds reported from this plant.
Comment: Table 4 needs IC50 for general comparison.
Response: We do agree that IC50 is also a good way to represent data. However, in the present study, the enzyme inhibitory activities of the extracts were evaluated as equivalents of standard inhibitors per gram of the plant extract (galantamine for acetylcholinesterase and butyrylcholinesterase, kojic acid for tyrosinase, and acarbose for α-amylase and α glucosidase inhibition assays). The way we have presented the data is very common and increasingly being accepted by the scientific community as evidenced by previous publications listed hereunder. In this perspective, our results tend to align with previous peer reviewed published scientific literature and we humbly request you to accept this format. (Zengin, G.., et al (2019), Food and Chemical Toxicology, 127, 237-250; Zengin G., et al., (2019), Industrial Crops and Products, 135, 107-121, Uysal, S., et al., (2018),. Food and Chemical Toxicology, 111, 525-536.; Zengin, G., et al., (2018). Food and Chemical Toxicology, 111, 423-431.; Zengin, G., et al., (2017). Food and Chemical Toxicology, 107, 540-553).
Therefore, we are making a plea to accept these data and grant us an opportunity to share these data to the scientific community which could subsequently open new perspectives for research.
Comment: Plant binominal name should be in italic
Response: We have ensured that the plant binominal name is in italic throughout.
Comment: Many typos throughout the manuscript.
Response: We have checked the manuscript thoroughly for typos and grammatical mistakes.

Reviewer 2 Report
- Please clarify how the 8 compounds for the docking studies were selected among the 50 ones.
- Please discuss on the ligand-protein interactions.
- Describe in brief the antioxidant and enzyme inhibition assays and cite the correct primary references.
- The species names should be given in italic.
Author Response
All the changes as indicated by the reviewer have been made in the revised manuscript.
Comment 1: Please clarify how the 8 compounds for the docking studies were selected among the 50 ones.
Response: These 8 compounds were selected based on their presence in the Heliotropium species.
Comment 2: Please discuss on the ligand-protein interactions.
Response: We have listed the interactions in new table (Table 6) and provided a few lines to discuss the interactions.
Comment 3: Describe in brief the antioxidant and enzyme inhibition assays and cite the correct primary references.
Response: We have provided the details of antioxidant and enzyme inhibition assays in the supplementary section.
Comment 4: The species names should be given in italic.
Response: We have made the species name italic throughout the manuscript.

Round 2
Reviewer 1 Report
Authors answered all the comments.
Reviewer 2 Report
I accept the corrections made by the authors.